# Research on Postcuring Parameters Effect on the Properties of Fiberglass-Reinforced Silicone Resin Coil Bobbin

**DOI:** 10.3390/ma16072588

**Published:** 2023-03-24

**Authors:** Hongmei He, Qiqi He, Hongchen Gao, Wei Hu, Song Xue

**Affiliations:** School of Manufacturing Science and Engineering, Southwest University of Science and Technology, Mianyang 621010, China

**Keywords:** FRSR, postcuring, thermostability, mechanical properties, compression molding

## Abstract

With the growing demand for insulation parts in extreme service environments, such as nuclear power, aviation, and other related fields, fiberglass-reinforced silicone resin (FRSR) has become a popular choice due to its exceptional physical and chemical properties in high-temperature and electromagnetic working environments. To enhance the performance of FRSR molded parts that can adapt to more demanding extreme environments, the oven postcuring process parameters on thermal stability and mechanical properties of the bobbin were investigated. The curing behavior of FRSR was analyzed by using thermogravimetric analysis (TGA) and the differential scanning calorimetry (DSC) method, and the bobbins were manufactured based on the testing results. Subsequently, the bobbins were oven postcured at different conditions, and the heat resistance and mechanical properties were analyzed by TGA and tensile tests. The results revealed that the tensile strength of the bobbin increased by 122%, and the weight loss decreased by 0.79% at 350 °C after baking at 175 °C for 24 h. The optimal process parameters for producing bobbins to meet the criteria of nuclear installations were determined to be a molding temperature of 120 °C, molding pressure of 50 MPa, pressure holding time of 3 min, oven postcuring temperature of 175 °C, and postcuring time of 24 h. The molded products have passed the thermal aging performance test of nuclear power units.

## 1. Introduction

With the rapid development of nuclear power, rockets, aerospace, radio, and electrical industries, the extreme working environment has higher and higher requirements for the working temperature, electrical insulation and lifespan of insulating parts [1,2,3,4]. The machinery also puts forward higher requirements for the complexity, surface quality, and dimensional accuracy of insulating parts. The materials that meet these requirements mainly include silicone resin and polyether ether ketone (PEEK) [5]. Silicone resins are usually formulated to contain a three-dimensional network of siloxane bonds, which are best known for their extreme temperature properties and have good chemical and electrical properties [6]. The silicone molding compound is mixed with silicone resin, filler and curing agent [7]. Silicone resin molding compounds are mixed with silicone resin, filler, catalyst, and curing agent [6]. As an engineering plastic, this material has high arc resistance, dielectric constant and power consumption coefficient, good corona resistance, and electric breakdown resistance. In addition, silicone resin can maintain a low coefficient of thermal expansion and water absorption in high-temperature and high-humidity environments [8].

Silicone molding compounds have low molding shrinkage, good high-temperature fluidity, and a high thermal-decomposing temperature. Therefore, the compression molding method is one of the general techniques for manufacturing silicone molding compounds, which is highly compatible with producing high-performance complex parts [9,10,11]. Silicone molding compound compression molding is used to manufacture insulating parts that can work for a long time above 200 °C, such as motor contactors and switchboards [12,13,14,15]. The compression molding process parameters of silicone molding compounds are one of the main factors that affect the performance and behaviors of the composites. The appropriate parameter is a crucial consideration for producing a matrix with a high degree of curing and eventually soak the fiber sheaf to provide better properties and performance of the silicone molding compound. Therefore, determination of the relationship between the compression molding process parameters of the silicone resin molding compound and the material properties is vital for producing competitive products.

Barone et al. [16] researched the process temperature effect on tensile properties of fiber-reinforced polyethylene composites. They found that specimens obtained at a molding temperature of 160 °C show the best performance in terms of tensile strength and modulus. Nevertheless, the lowest values of strength and modulus obtained by using specimens processed at a molding temperature of 220 °C. For the surface with pores and pits on the specimens at 220 °C, they attributed it to fiber degradation [16]. Takagi Asano [17] found that different pressures parameters applied in compression molding would affect the bending strength of cellulose microfiber nanofiber-reinforced polymer. The bending strength of composites is greatly affected by the forming pressure, and the bending strength increases with the increase of the forming pressure. However, it almost reaches the limit under the molding pressure of 50 MPa. This condition indicates that the pressure used in the molding process has the best value when producing composite materials with competitive properties. Medina et al. [18] revealed that the pressure applied in the compression molding process has a substantial impact on the mechanical properties of natural fiber composites, and explained the best pressure measurement used in the manufacture of kenaf epoxy composites. Note that 6 MPa is an important pressure for the manufacture of kenaf epoxy composites. Higher pressure increments lead to fiber structure damage and lower bending strength of the composites. Mat Khandar and Akil studied the relationship between the forming conditions and the impact resistance of flax-reinforced PLA composites. Their research showed that the material achieves maximum impact strength under the condition of molding temperature of 200 °C, the molding pressure of 30 bar and maintained for 3 min. [19]. Verma et al. [20] also ascribe the influence of packing time in compression molding to the bending property of short fiber-reinforced resin composites. In addition, their study showed that the bending property increased with the increase of packing time, by up to 10 min [21]. Meanwhile, the study by Satoshi and Keita indicated the tensile strength of the hemp-reinforced PLA composites effect by mold holding time. The tensile strength value of hemp PLA composites goes up with the increase of mold holding time [22,23]. After compression molding, thermosetting resin needs postheat treatment to cure it completely. The strength parameters and thermal deformation temperature of glass fiber-reinforced resin can be significantly enhanced by a reasonable oven postcuring process. Park et al. [24] explored the influence of the oven postcuring process on the mechanics of glass aluminum reinforced epoxy resin (GLARE) laminates. After oven postcuring, the void content of glass aluminum-reinforced epoxy decreased from 1.51% to 0.89%, which can increase the interlaminar shear strength of glass aluminum-reinforced epoxy resin laminate by about 3.6%. The research of Deringer et al. [25] found that the mechanical properties of the epoxy-based compression molding component improved relative to the degree of curing time, and the full curing time is 600 s.

In general, the material properties of the compressed components fabricated by FRSR were related to the degree of curing significantly. To enhance the degree of curing of the compressed components, many manufacturers used the trial-and-error method at the start of manufacture new parts. However, the trial-and-error method has no specific reference basis and usually increases the final production cost. Manufacturing based on the curing behavior of FRSR and the relationship between process parameters and material properties is an efficient method in the improvement of production quality and reducing production costs. In this study, thermogravimetric analysis (TGA) and differential scanning calorimetry (DSC) were used to analyze the curing behavior of FRSR. TGA and tensile tests were used to study the effect of oven postcuring temperature and time on the thermal stability and mechanical properties of FRSR. In addition, access tests were carried out to verify that the coil bobbin produced by the process parameters obtained by the thermal test meets the nuclear installations standards.

## 2. Materials and Experiments

### 2.1. Materials

As shown in Figure 1, the silicone molding compound used in this study is a commercial product. The material is a long, fiberglass-reinforced silicone resin molding compound, which is mainly made of silicone resin, glass fiber, colorants and other fillers by precision compression molding. It can normally maintain good insulation performance in a wide temperature and frequency range, under 4.9 kV/mm of the dielectric strength, 189 s of the arc resistance, 4 of the dielectric constant. The specific performance indicators of this material are shown in Table 1. It meets the requirements of ASTM D5948 Type MSI-30 [26].

### 2.2. Thermal Analysis Test for Materials

TGA was carried out on 5 mg specimens by using a TA Q500 thermogravimeter (TA Instruments, New Castle, DE, USA) under oxygen flow at 10 °C/min from 20 °C to 620 °C. The temperature of the beginning of the inflection point and the maximum weight loss rate were considered as the thermal degradation temperature.

DSC test were carried out on a sample of 5 mg in aluminum pans by using a TA Q200 differential scanning calorimeter under oxygen atmosphere where the scanning rate is 10 °C/min, heated from 25 °C to 275 °C. The endothermic temperature, exothermic temperature, and glass transition temperatures (TGA) of the material were determined by DSC.

### 2.3. Experimental Procedures

The dimension of the magnet yoke coil bobbin is 100 mm in outer diameter, 70 mm in height, and 6 mm in maximum thickness. As shown in Figure 2, using a close-up-type stainless steel mold, an FRSR coil bobbin was manufactured by compression molding in a four-column hydraulic press Y32-315. First, 227.6 g of compound was weighed and added to the cavity of a compression mold installed on the four-column hydraulic press. According to the process parameters recommended by the supplier of the material, as shown in Table 2, the coil bobbin was molded under the process parameters of a molding temperature of 145 °C, molding pressure of 50 MPa, and holding time of 2 min. Finally, the compression mold was opened by the hydraulic press and the coil bobbin was taken out. As shown in Figure 3a, the surface of the coil bobbin is rough, and there are many cracks in the cylindrical part. According to the DSC test of SI-9041A molding material compound, the initial temperature of exothermic peak, the temperature of exothermic peak, and the end temperature of exothermic peak are 130 °C, 179 °C, and 230 °C, respectively. Considering the exothermic situation of the curing reaction in the molding process and the influence of the heating rate on the peak temperature, the faster the heating rate, the higher the initial temperature of the reaction peak. Therefore, the mold temperature of 125 °C to 140 °C makes the mold plastic melt and at the same time promotes the uniform mixing and filling of the material as far as possible, ensures the molding of the product structure, prolongs the pressure holding time, and ensures the curing of the material. After many experiments, the final coil bobbin is as shown in Figure 3b. The material was compressed by mold at a molding temperature of 125 °C and a molding pressure of 50 MPa for 3 min, which obtained a coil bobbin with good quality in terms of appearance. According to the DSC test results, the postcuring temperature of the coil bobbin is about 179 °C. According to the TGA test results, it is found that the curing weight loss temperature range of raw materials is mainly in the range of 100°C–215°C. Theoretically, the postcuring temperature should be higher than the curing reaction temperature, so the baking temperature was set to 135 °C, 175 °C, and 215 °C, and the baking time was set to 5 h and 24 h. Qualified coil bobbins were deburred and put into an oven for postcuring. This step was performed in a forced-draft oven with good temperature control and outside venting. As shown in Table 3, the baking temperature was set to 135 °C, 175 °C, and 215 °C, and the baking time was set to 5 h and 24 h. The coil bobbins after oven postcuring are shown in Figure 4.

### 2.4. Thermogravimetric Analysis (TGA)

The FRSR coil bobbins after oven postcuring in different temperatures and times were, respectively, subjected to thermogravimetric tests. TGA was performed on a 5 mg coil bobbin sample by using TA Q500 thermogravimetry under oxygen flow at a rate of 10 °C/min from 20 °C to 620 °C. The thermal degradation temperature considered is the weight loss rate at the same temperature.

### 2.5. Tensile Properties

Tensile testing was carried out at 25 °C with a constant rate of 5 mm/min tested with an electronic tensile testing machine CMT5305 according to the ASTM D638-14 [27]. In order to study properties of FRSR in the actual structure, FRSR coil bobbins were cut into 80 (mm) × 16 (mm) × 8 (mm) tensile test specimens by using a numerical control router. Specimens were, respectively, taken from the upper flange, the middle cylinder, and the lower flange of the coil bobbins. All results were taken from the average value of three samples.

## 3. Results and Discussion

### 3.1. Thermal Analysis Test for Materials

For the molding of thermoset materials, the curing behavior of raw materials is considered to be one of the critical factors in the molding process. TGA and DTG curves for FRSR molding compound were shown in Figure 5. As shown in the TGA curve (Figure 5a), the sample presents five mass loss regions which are located around 60–100 °C, 100–215 °C, 215–350 °C, 350–540 °C, and 540–620 °C. The first weight-loss zone below 100 °C is generally considered to be caused by the evaporation of surface water in the sample while the 100–215 °C regions should be associated with the curing of silicone resin. When the temperature range is between 215–350 °C, the cured silicone resin has good thermal stability. The weight loss of the sample began to increase significantly above 350 °C, which was caused by the thermal aging decomposition of the silicone resin. Below 620 °C, the residual amount of the sample was as high as 92%. The main components of the residual amount of the sample were silicone resin and glass fiber.

The DTG curve in Figure 5b shows that the maximum weight loss rate (Tmax) of the material curing occurred at 150 °C. Within the range of 350–540 °C, the silicone resin begins to undergo thermal-oxidative degradation in an air atmosphere, and the organic groups attached to the silicon atoms begin oxidation removal [28]. The decomposition peak with a maximum weight loss rate (Tmax) of 430 °C corresponds to the “bite-back” reaction of the hydroxyl group at one end of the main molecular chain in the silicone resin, which triggers the “unbutton” degradation of the main molecular chain [29,30]. In the range of 540–620 °C, the silicone resin also has a peak with the largest decomposition rate at approximately 560 °C, which is due to the breaking and rearrangement reaction of the main molecular chain Si–O–Si bond [29,30]. These results are in accordance with other studies of thermal aging of silicone resin [31]. DSC curves for FRSR molding compound is shown in Figure 6. The heating scans show the glass transition temperature (TGA) of the silicone resin in samples at about 89 °C. The samples exhibit an endothermic peak at about 123 °C. This was related to the melting of colorants and other fillers in the sample. Another peak is a main exothermic peak located around 179 °C corresponding to the curing of silicone resin.

### 3.2. Thermogravimetric Analysis (TGA) of the Samples after Postcuring Treatment

The thermal stability of coil bobbins under different postcuring process conditions was analyzed by thermogravimetric analysis test. Each set of trials was repeated three times. Figure 7 showed the TGA curve of the coil bobbin after baking at different temperatures for 5 h and 24 h. For the coil bobbin, the thermal weight loss process was mainly divided into two stages. The first stage of thermal weight loss was related to the curing degree of the coil bobbin. The uncured silicone resin contained in the coil bobbin continued to undergo a curing reaction between 130 °C and 200 °C. The second stage was caused by the thermal-oxidative degradation of the silicone resin. This process, starting at around 350 °C, was consistent with the thermal weight loss curve of the FRSR molding compound after 350 °C. The unbaked coil bobbin weight loss rate is the highest at 350 °C, followed by 135 °C and 215 °C and is the lowest at 175 °C. The results show that the baking treatment improves the thermal stability of the coil bobbin. This is because the uncured silicone resin material in the coil bobbin continues the curing reaction at an appropriate temperature, which increases the cross-linking density of the resin and improves the thermal weight loss of the product. It can also be seen in Figure 7 that the coil bobbin baked at 175 °C hardly underwent the first stage of thermal weight loss because it has the highest degree of curing and the optimal thermal stability. This result corresponds to the exothermic peak in the DTG curve of the FRSR molding compounds.

After baking at the same temperature for 5 h and 24 h, the TGA curve of the coil bobbin is shown in Figure 8. The weight loss rate of the coil bobbin after baking for 24 h was slightly lower than that of the coil bobbin after baking for 5 h. It indicates that extending the baking time can slightly promote the curing reaction acceleration of the silicone resin, thereby improving the thermal stability of the coil bobbin, but the effect of this method is limited.

### 3.3. Tensile Properties

The influence of different oven postcuring temperatures and times on the tensile properties of the FRSR coil bobbin were studied. The sample after the tensile test is shown in Figure 9. The tensile behavior of the coil bobbin samples shown in Figure 10 gives the relationship between tensile stress and strain at break from 135 °C, 175 °C, and 215 °C baked and unbaked samples. The tensile strength (TS) of the baked samples was improved compared to that of unbaked samples. When the baking temperature was 175 °C, the TS of the sample increased the most. When the baking temperature was 215 °C, the TS of the sample increased the least. Compared with non-poscuring samples, the TS of the coil bobbin is increased by 122% when the curing temperature is 175 °C and the curing time is 24 h.

The results show that baking at the right temperature improves the tensile properties of the FRSR coil bobbin. Combined with the DTG curve of the FRSR molding compound, this implies that the uncured silicone resin has a faster heat release rate under an environment of 175 °C. Baking accelerates the curing reaction rate of the silicone resin in the coil bobbin and increased strength of its curing degree. In addition, it is found that the coil bobbin baked at 215 °C has a higher curing degree than that baked at 135 °C, but with lower tensile properties. According to the literature [30], this is caused by the thermal stress inside the coil bobbin due to extensively high baking temperature, thus resulting in low tensile properties of the samples.

After baking for 5 h and 24 h at the same temperature, the tensile properties of the coil bobbin samples are shown in Figure 11. When the baking time was extended to 24 h at 175 °C, the (TS of the samples was increased by 26%. At 135 °C, the TS of the samples ascended by 13%. At 215 °C, the TS of the samples improved by 9%. From these results, it is clear that extending the baking time can further increase TS of the samples. The mechanical properties of thermosetting materials are more sensitive to changes of curing degree than thermal stability, and the mechanism has been studied by many authors [24]. Therefore, with an increase of the baking time, the curing degree of the silicone resin in the FRSR coil bobbin increases slightly while the TS of the sample increases significantly. At this time, the slight change of curing degree greatly improves the mechanical properties of the overall coil bobbin.

### 3.4. Access Testing

According to the regulations of the nuclear installations acceptance tests [32], the FRSR coil bobbins after the encapsulation were subjected to a continuous heat exposure test at 310 °C for 7 days. The thermal aging test was carried out on the coil tube obtained under the optimum molding process (curing temperature 175 °C, curing time 24 h). Figure 12 reflects the 7-day heat exposure results of the coil assemblies at 310 °C, in which the coil bobbin has no obvious carbonization. After the heat exposure was completed, a series of engineering verification tests were carried out on the material properties of the coil assembly according to actual engineering requirements [32]. In a vibrating environment, an insulation resistance test was carried out. The inductance test and 500 V pulse voltage test were carried out in a damp environment. A 1000 V pulse voltage test, a withstand voltage test and a coil resistance test were carried out as well. Test results are shown in Table 4. The appearance and material properties of the coil assembly meet the nuclear installations standards, and the results have also been verified in the engineering application of the project collaborating company.

## 4. Conclusions

This paper has studied the relationship between FRSR molding product properties and the oven postcuring parameters of compression molding. In this study, TGA and DSC of the FRSR molding compound were used to analyze the curing behavior of the material. The FRSR coil bobbins were successfully prepared to use the compression molding process. The oven postcuring process of the FRSR coil bobbin was studied to analyze the effect of the baking temperature and the baking time on the properties of the coil bobbin. The thermal stability and tensile properties were employed to evaluate the quality of the coil bobbin. The main conclusions drawn by this study are as follows.
(1)The TGA of the FRSR molding compound is 89 °C. The maximum weight loss during the curing occurs at 150 °C, and the material emits the greatest amount of heat at 179 °C.(2)FRSR coil bobbins of fiberglass-reinforced composite baked at a suitable temperature and time were equipped with higher thermal stability and strength compared with unbaked coil bobbins. The thermal stability and tensile properties of coil bobbins increase first and then decrease with the increase of the baking temperature. Baking at 175 °C, coil bobbins have higher thermal stability and tensile properties. Extending the baking time can improve the strength and modulus of coil bobbins but limits the effect on increasing thermal stability.(3)The compression molding process parameters for the coil bobbin are as follows: the mold temperature of 120 °C, the molding pressure of 50 MPa, the pressure holding time of 3 min, the curing temperature of 175 °C, and the baking time of 24 h.

## Figures and Tables

**Figure 1 materials-16-02588-f001:**
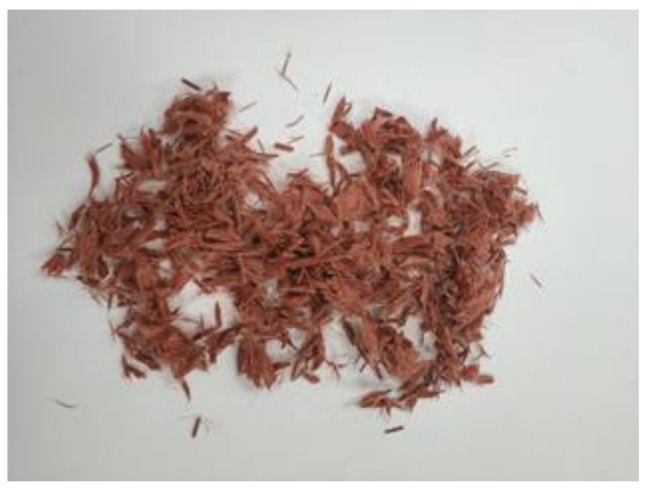
Material.

**Figure 2 materials-16-02588-f002:**
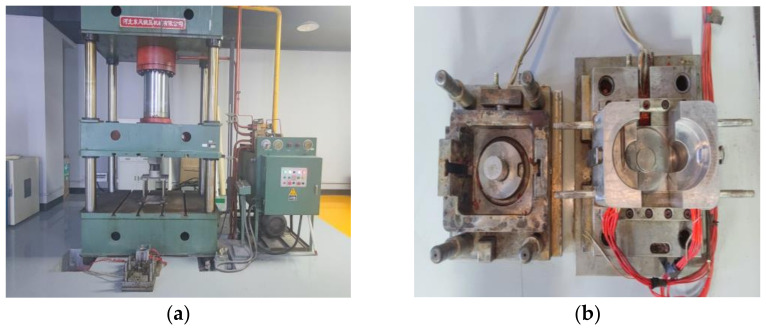
Experimental production equipment. (**a**) Hydraulic press. (**b**) Compression mold.

**Figure 3 materials-16-02588-f003:**
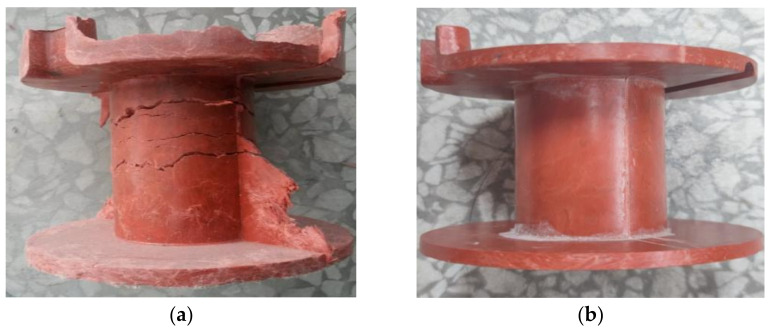
Compression-molded FRSR coil bobbin. (**a**) By process parameters recommended with the raw material supplier. (**b**) Process parameters optimized based thermal analysis tests.

**Figure 4 materials-16-02588-f004:**
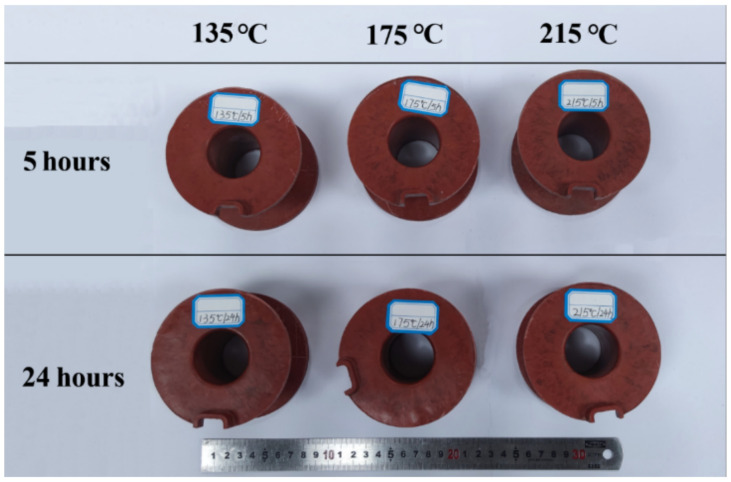
Silicone resin coil bobbins after oven postcuring.

**Figure 5 materials-16-02588-f005:**
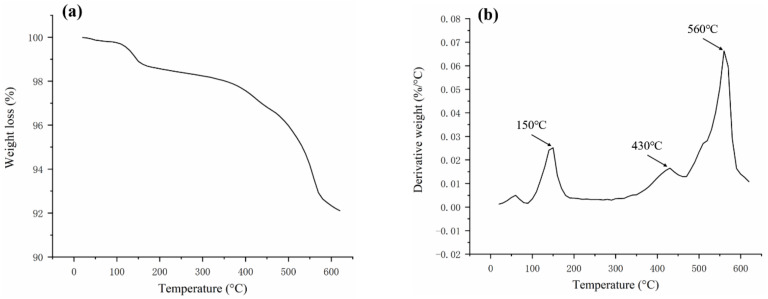
(**a**) TGA and (**b**) DTG curves of silicone molding compound at a heating rate of 10 °C/min.

**Figure 6 materials-16-02588-f006:**
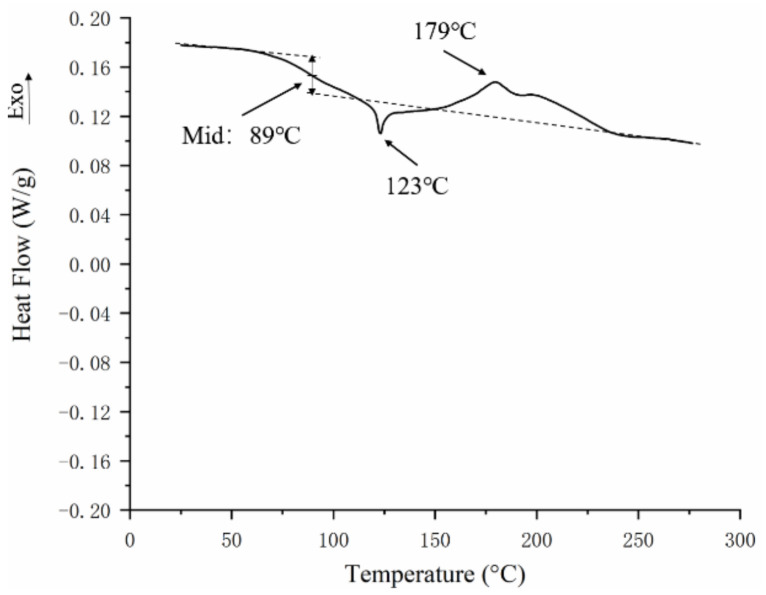
Heating DSC curves for silicone molding compound.

**Figure 7 materials-16-02588-f007:**
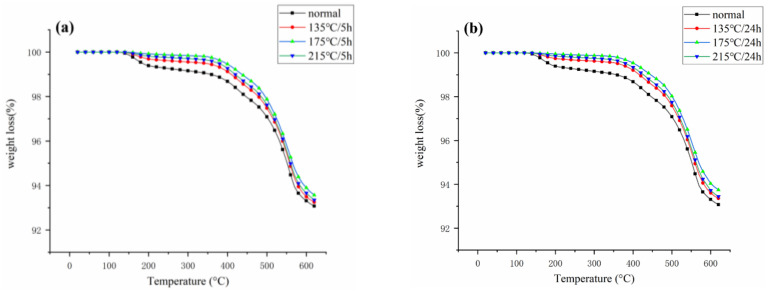
TGA curve of the coil bobbin after (**a**) 5 h and (**b**) 24 h of baking.

**Figure 8 materials-16-02588-f008:**
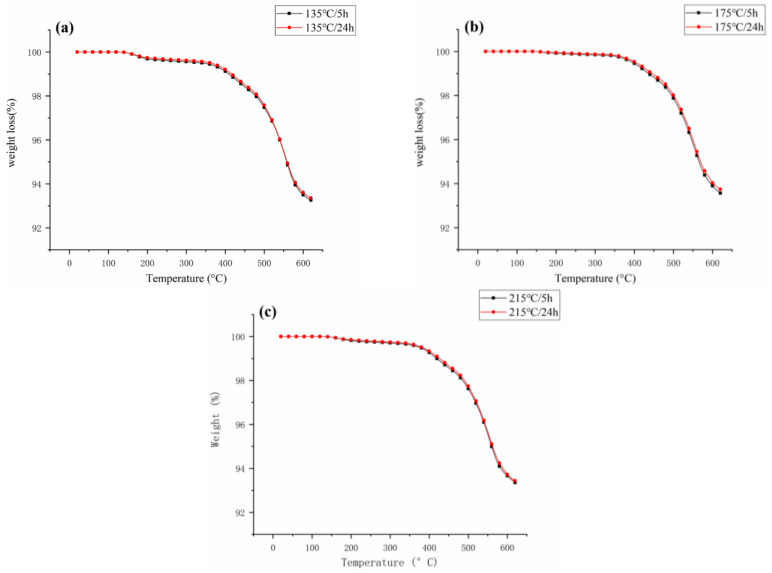
TGA curve of the coil bobbin after baking (**a**) 135 °C, (**b**) 175 °C, and (**c**) 215 °C.

**Figure 9 materials-16-02588-f009:**
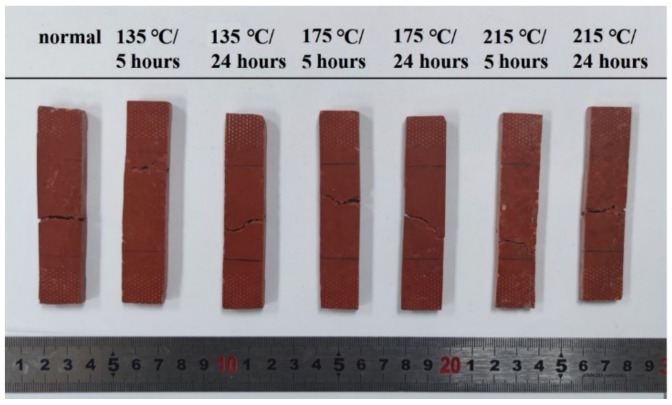
The appearance of the specimens after the tensile test.

**Figure 10 materials-16-02588-f010:**
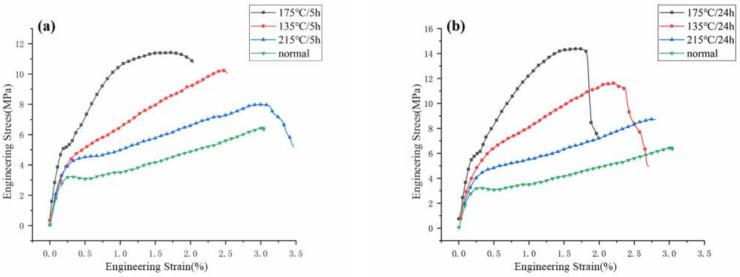
Average flexural stress vs. deformation of specimens after baking for (**a**) 5 h and (**b**) 24 h.

**Figure 11 materials-16-02588-f011:**
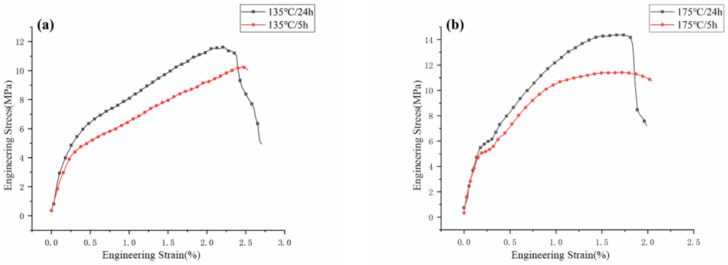
Average flexural stress vs. deformation of specimens after baking at (**a**) 135 °C, (**b**) 175 °C, and (**c**) 215 °C.

**Figure 12 materials-16-02588-f012:**
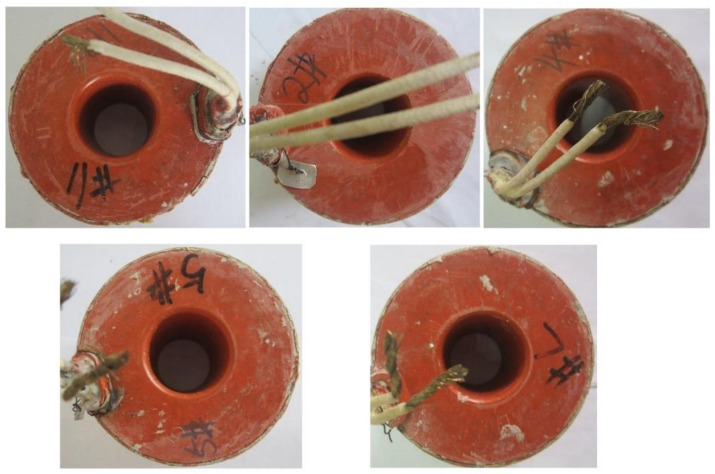
The coil assemblies after 7 days of heat exposure at 310 °C.

**Table 1 materials-16-02588-t001:** Material properties.

Properties	Typical Value	Unit	Method
Relative density	1.92	g/cm^3^	ASTM D792
Molding shrinkage	0.15–0.25	%	ASTM D955
Water absorption	0.24	% at 50 °C for 48 h	ASTM D570
Deflection under load postbaked	>282	°C at 1.8 MPa	ASTM D648A
Tensile stress at break	31	MPa	ASTM D638
Compressive strength	61	MPa	ASTM D695

**Table 2 materials-16-02588-t002:** Suppliers suggested start-up conditions.

Compression	Parameter Settings
Molding pressure—compression	13.6–54.4 MPa
Mold temperature	138–149 °C
Cure time	60–90 s (per 3.2 mm thickness)

**Table 3 materials-16-02588-t003:** The coil bobbin oven postcuring parameters.

Grouping	Temperature	Holding Time
1	Normal	Normal
2	135 °C	5 h
3	175 °C	5 h
4	215 °C	5 h
5	135 °C	24 h
6	175 °C	24 h
7	215 °C	24 h

**Table 4 materials-16-02588-t004:** The coil assemblies engineering verification results.

Numbering	Resistance to Ground 500 V/1 min Ω	Conductor Resistance Ω	Inductance mH	Impulse Voltage 500 V	Impulse Voltage 1000 V	Withstand Voltage 1500 V/1 min
Acceptance criteria	≥0.5 × 10^5^	1.239±3%	10–12			
No. 7	1.87 × 10^5^	1.2195	11.006	00/00	00/00	Pass
No. 11	1.46 × 10^5^	1.2238	11.031	00/01	00/00	Pass
No. 5	8.21 × 10^5^	1.2118	10.933	00/00	00/00	Pass
No. 2	4.71 × 10^5^	1.2252	11.002	00/01	00/00	Pass
No. 4	1.49 × 10^5^	1.2108	10.902	00/00	00/01	Pass
Min	1.46 × 10^5^	1.2108	10.902	NA	NA	Pass
Deviation	+0.96 × 10^5^	−0.0282	NA	NA	NA	Pass

## Data Availability

The raw/processed data required to reproduce these findings cannot be shared at this time due to technical or time limitations.

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
