# Peer review of "Research on Postcuring Parameters Effect on the Properties of Fiberglass-Reinforced Silicone Resin Coil Bobbin"

_materials, 2023, doi:10.3390/ma16072588_

Round 1
Reviewer 1 Report
The manuscript by He et al. describes the investigation of the thermal and mechanical properties of bobbins made of fiberglass-reinforced silicone resin (FRSR). The authors first examined the relationship between FRSR molding product properties and the oven post-curing parameters of compression molding. TG and DSC of the FRSR molding compound were used to analyze the curing behavior of the material. The FRSR coil bobbins were then prepared using the compression molding process. The oven post-curing process of the FRSR coil bobbin was analyzed to obtain the effect of the baking temperature and the baking time on the properties of the coil bobbin. The thermal stability and tensile properties were also studied to evaluate the quality of the coil bobbin.
In my opinion, the study was competently performed, and the conclusions are supported by the data. The authors just have to improve their English.
After this minor revision, I can recommend the acceptance of the manuscript.
Author Response
Thank you very much for your efforts.

Reviewer 2 Report
Report:
Fiberglass reinforced silicone resin (FRSR) is a type of composite material with excellent properties, particularly its ability to withstand extreme service environments. One major application for FRSR is to construct insulating parts for structures that serves under extreme environments. Such components are commonly manufactured by a ‘compression molding process’, during which the silicone molding compounds are compressed with a controlled pressure, at an elevated temperature, and with a certain period of holding time. After the compression processing, the component will then undergo a post-curing process in the oven, with controlled temperatures and time durations.
He et al. conducted an experimental study to investigate the impact of post-curing temperature and duration on the thermal stability and mechanical properties of as-compressed FRSR coil bobbins. The study used Differential Scanning Calorimetry (DSC) and Thermal Gravimetric Analysis (TGA) to guide for parameter selection.
The results indicate that thermal stability improves with increasing post-curing temperature, peaking at 175 degrees before decreasing. While increasing the duration of treatment does improve thermal stability, it is not as significant as increasing the temperature. Regarding mechanical properties, the best performance was observed in the group of samples with the best thermal stability, achieved at 175 degrees for 24 hours. The impacts of post-cruing parameters, and the set of optimised parameters for this composite, are the most important knowledge output for this work, which I agree. However, there are quite some issues in the current version of the manuscript, which should be resolved before considering for publication in this journal.
- The authors claims that their approach is different from a ‘trial-an-error’ method (Line 91), likely due to the approach they used is referenced by TGA and DSC results (Line 95). However, TGA in this work was basically used to measure the thermal stability by evaluating the weight loss. DSC could be more informative as the peak observed at 179 degree is likely to be the reason for choosing 175 degree for postcuring processing, but the reason was not given. Anyway, the analysis on the DSC results and how it can be used as a reference is not clear.
- In the paragraph 2 starting from line 55. The paragraph reviews the effects of several processing parameters on the properties, which is great. However, the content related to the main focus of this paper, which is the effect of post-curing parameters, are not sufficient (starting from Line 80).
- Information about the changes that the composite undergoes under post-curing was not provided in the Introduction. Several concepts in the paragraph starting from Line 182, such as ‘Oxidation removal’, ‘Unbutton’ degradation, should be introduced.
- Line 95, aberrations ‘TGA’ and ‘DSC’ are not provided with their full terms. This should be done at the first time they show up in the text.
- Line 132, is ‘TG’ a typo for ‘TGA’?
- In Figure 9, non-standard tensile tests samples were used for mechanical testing. How will this affect the mechanical test results?
- In Figure 11, all stress–strain curves should have x-axis titled ‘engineering strain’ and y-axis titled ‘engineering stress’.
- Providing the reason why 175 is the optimised temperature will make Conclusion 3 more significant.
- There is too many language issues, please go though a careful language check. I listed a few as follows:
- There should be a space beteen the number and its unit. E.g. Line 118 5mg should be 5 mg, Line 130 ‘50Mpa’ should be 50 ‘MPa’.
- There is a lot of typos. E.g. Line 82 ‘posturing’, Line 126 ‘conpound’,
- Extra space in Line 215
- Line 122, title of section 2.3 ‘Experimental part preparation’ is better to be ‘Experimental procedures’?
Author Response
Thank you very much for your efforts, our respons as follow:
Reviewer 2:
Fiberglass reinforced silicone resin (FRSR) is a type of composite material with excellent properties, particularly its ability to withstand extreme service environments. One major application for FRSR is to construct insulating parts for structures that serves under extreme environments. Such components are commonly manufactured by a ‘compression molding process’, during which the silicone molding compounds are compressed with a controlled pressure, at an elevated temperature, and with a certain period of holding time. After the compression processing, the component will then undergo a post-curing process in the oven, with controlled temperatures and time durations.
He et al. conducted an experimental study to investigate the impact of post-curing temperature and duration on the thermal stability and mechanical properties of as-compressed FRSR coil bobbins. The study used Differential Scanning Calorimetry (DSC) and Thermal Gravimetric Analysis (TGA) to guide for parameter selection.
The results indicate that thermal stability improves with increasing post-curing temperature, peaking at 175 degrees before decreasing. While increasing the duration of treatment does improve thermal stability, it is not as significant as increasing the temperature. Regarding mechanical properties, the best performance was observed in the group of samples with the best thermal stability, achieved at 175 degrees for 24 hours. The impacts of post-cruing parameters, and the set of optimised parameters for this composite, are the most important knowledge output for this work, which I agree. However, there are quite some issues in the current version of the manuscript, which should be resolved before considering for publication in this journal.
- The authors claims that their approach is different from a ‘trial-an-error’ method (Line 91), likely due to the approach they used is referenced by TGA and DSC results (Line 95). However, TGA in this work was basically used to measure the thermal stability by evaluating the weight loss. DSC could be more informative as the peak observed at 179 degree is likely to be the reason for choosing 175 degree for postcuring processing, but the reason was not given. Anyway, the analysis on the DSC results and how it can be used as a reference is not clear.
R: (Lines 149-154): Thanks. We have already added correlation analysis in the manuscript. In practical application, the initial reaction temperature, maximum reaction temperature and end reaction temperature of the exothermic peak of DSC curve correspond to the approximate gel temperature, curing temperature and post-curing temperature of the material, respectively. And according to the TGA test results, it is found that the curing weight loss temperature range of raw materials is mainly in the range of 100°C-215°C. Theoretically, the post-curing temperature should be higher than the curing reaction temperature, which is why we set 135°C,175°C and 215 as the post-curing temperature of the coil frame.
- In the paragraph 2 starting from line 55. The paragraph reviews the effects of several processing parameters on the properties, which is great. However, the content related to the main focus of this paper, which is the effect of post-curing parameters, are not sufficient (starting from Line 80).
R: (Lines 86-89): Thank you for your suggestion. We have added a description about post-curing of materials in the manuscript.
- Information about the changes that the composite undergoes under post-curing was not provided in the Introduction. Several concepts in the paragraph starting from Line 182, such as ‘Oxidation removal’, ‘Unbutton’ degradation, should be introduced.
R: In this study, we focused on the forming process and post-curing parameters effect on the performance of the product. And restrict to our research background, the detail of the curing mechanism is not the research emphasis.
- Line 95, aberrations ‘TGA’ and ‘DSC’ are not provided with their full terms. This should be done at the first time they show up in the text.
R: (Lines 99-100). Thanks. We have corrected it in the appropriate position of the manuscript.
- Line 132, is ‘TG’ a typo for ‘TGA’?
R: Thank you for your question. We have corrected it in the appropriate position of the manuscript.
- In Figure 9, non-standard tensile tests samples were used for mechanical testing. How will this affect the mechanical test results?
R: Due to the limitation of the structure and size of the coil framework, we cannot cut the standard sample. However, as can be seen from Figure 9, the gauge distance at both ends of the sample is divided according to the standard size to ensure the accuracy of the test results as much as possible. Moreover, from the results of tensile tests, the existence of non-standard part errors will not affect the overall trend. The conclusion drawn from the tensile test is still reliable.
- In Figure 11, all stress–strain curves should have x-axis titled ‘engineering strain’ and y-axis titled ‘engineering stress’.
R: Thank you for your advice. We have corrected these pictures.
- Providing the reason why 175 is the optimised temperature will make Conclusion 3 more significant.
R: (Lines 149-154): We have added the reason for choosing 175° as the post-curing temperature
- There is too many language issues, please go though a careful language check. I listed a few as follows:
- There should be a space beteen the number and its unit. E.g. Line 118 5mg should be 5 mg, Line 130 ‘50Mpa’ should be 50 ‘MPa’.
- There is a lot of typos. E.g. Line 82 ‘posturing’, Line 126 ‘conpound’,
- Extra space in Line 215
- Line 122, title of section 2.3 ‘Experimental part preparation’ is better to be ‘Experimental procedures’?
R: Thank you for your suggestion, we have to correct the above problems, and to correct the language errors in the manuscript has carried on the inspection

Reviewer 3 Report
The authors described the post-curing thermal and mechanical properties of fiberglass reinforced silicone resin coil bobbin as insulating products for nuclear applications. The authors present interesting results for researchers that activate in the field of materials exposed to extreme conditions. However, I recommend publication after taking into account the following suggestions:
Point 1: Lines 35-41: The sentences between these lines should be rephrased in the same paragraph. They all refer to the same silicon molding compounds so no need to segregate their properties in three different sentences.
Point 2: Lines 131: Please rephrase this sentence “As shown in Figure 3a, according to the problems on the coil bobbin and the DSC and TG results, 132 the molding process parameters were step by step adjusted to made again”
Point 3: Please change the title of 3.2 Thermogravimetric analysis (TGA) to a more specific step of your work for instance “Thermogravimetric analysis (TGA) of the samples after thermal treatment” or post-curing treatment or after baking.
Point 4: Lines 203-204: Please rephrase “In order to study the influence of different oven postcuring temperatures and times 203 on the thermal stability of the FRSR coil bobbin. TG curves were used to determine the 204 thermal stability of the coil bobbins.”
Point 5: Line 215: Please rephrase “This is because that the curing reaction speed of uncured silicone resin part in the coil 215 bobbin accelerates at the right temperature.”
Point 6: The legend of Figure 8 seems ambiguous and needs changing. Please refer to the graphs from Figure 8 as TG comparative results of coil bobbin at 135, 175 and 215 after baking for 5, respectively 24 hours. Also, the text from the manuscript should describe more thoroughly the graphs from Figure 8.
Point 7: The results exposed in Figure 8 reveal a slight change in weight loss after baking to 5 or 24 hours. Did the authors try to repeat their tests? Please specify in this section how many tests were performed on the same samples in terms of TG analysis.
Point 8: In the abstract the authors claim the improvement in tensile strength with 122% of the baked samples at 175 °C for 24 hours, but in Section 3.3 this result is not discussed.
Point 9: Legend from Figure 11 should be modified as mentioned before for Figure 8, as comparative results between the samples baked at different temperatures for 5, respectively 24 hours.
Point 10: The description of the samples presented in Section 3.4 is not clear. The samples exposed at 310 C were baked for 5, or 24 hours at different temperatures or these tests were made only for blank samples? Please give a more detailed description of your tests.
Point 11: In Table 4 standard deviation should be added. Also, the authors should give more details about the importance of these tests from Section 3.4 and the results that they obtained.
Point 12: In the Abstract or Conclusion Section the authors din not mentioned anything about the results from section 3.4.
Author Response
Thank you for your efforts,our respons as follow:
Reviewer 3:
The authors described the post-curing thermal and mechanical properties of fiberglass reinforced silicone resin coil bobbin as insulating products for nuclear applications. The authors present interesting results for researchers that activate in the field of materials exposed to extreme conditions. However, I recommend publication after taking into account the following suggestions:
Point 1: Lines 35-41: The sentences between these lines should be rephrased in the same paragraph. They all refer to the same silicon molding compounds so no need to segregate their properties in three different sentences.
R: (Lines 36-42): Thanks for the suggestion. We have reorganized the wording of these three sentences to make the manuscript more logical.
Point 2: Lines 131: Please rephrase this sentence “As shown in Figure 3a, according to the problems on the coil bobbin and the DSC and TG results, 132 the molding process parameters were step by step adjusted to made again”
R: (Lines 137-146):Thank you for your suggestion. We have rewritten this part and added more details.
Point 3: Please change the title of 3.2 Thermogravimetric analysis (TGA) to a more specific step of your work for instance “Thermogravimetric analysis (TGA) of the samples after thermal treatment” or post-curing treatment or after baking.
R: (Line 217): We have corrected it.
Point 4: Lines 203-204: Please rephrase “In order to study the influence of different oven postcuring temperatures and times 203 on the thermal stability of the FRSR coil bobbin. TG curves were used to determine the 204 thermal stability of the coil bobbins.”
R: (Lines 219-220): We have corrected it.
Point 5: Line 215: Please rephrase “This is because that the curing reaction speed of uncured silicone resin part in the coil 215 bobbin accelerates at the right temperature.”
R: (Lines 229-232): We have corrected it.
Point 6: The legend of Figure 8 seems ambiguous and needs changing. Please refer to the graphs from Figure 8 as TG comparative results of coil bobbin at 135, 175 and 215 after baking for 5, respectively 24 hours. Also, the text from the manuscript should describe more thoroughly the graphs from Figure 8.
R: Figure 8 is a thermogravimetric curve of several groups of molded products after curing for different periods of time at the same temperature. From these figures, it can be analyzed that extending the curing time can improve the thermal stability of the product, but the improvement effect is not obvious.
Point 7: The results exposed in Figure 8 reveal a slight change in weight loss after baking to 5 or 24 hours. Did the authors try to repeat their tests? Please specify in this section how many tests were performed on the same samples in terms of TG analysis.
R:(Lines 220-221): Thank you for your suggestion. We repeated three sets of trials to ensure reliable results. It also gives a supplementary explanation in the paper.
Point 8: In the abstract the authors claim the improvement in tensile strength with 122% of the baked samples at 175 °C for 24 hours, but in Section 3.3 this result is not discussed.
R: (Lines 255-257): Thank you for your suggestion. We have made a supplementary explanation.
Point 9: Legend from Figure 11 should be modified as mentioned before for Figure 8, as comparative results between the samples baked at different temperatures for 5, respectively 24 hours.
R: Figure 11 shows the tensile curves of several molded products after curing for different periods of time at the same temperature. From these pictures, it can be analyzed that extending the curing time can greatly improve the tensile strength of the product.
Point 10: The description of the samples presented in Section 3.4 is not clear. The samples exposed at 310 C were baked for 5, or 24 hours at different temperatures or these tests were made only for blank samples? Please give a more detailed description of your tests.
R: (Lines 292-293): Thank you for your suggestion. We have explained it in the manuscript.
Point 11: In Table 4 standard deviation should be added. Also, the authors should give more details about the importance of these tests from Section 3.4 and the results that they obtained.
R: (Line 307): Thanks for your suggestion, we have modified Table 4.
Point 12: In the Abstract or Conclusion Section the authors din not mentioned anything about the results from section 3.4.
R: (Lines 22-23): Thanks for your suggestion, we've added that in the abstract section.

Round 2
Reviewer 2 Report
The authors did a good job on the revision. Now I suggest accepting the manuscript in its current form.
Reviewer 3 Report
Dear All,
I recommend the publication of your manuscript considering that you have changed the manuscript accordingly to my suggestions.
Kind regards